# Germline Mutations in Steroid Metabolizing Enzymes: A Focus on Steroid Transforming Aldo-Keto Reductases

**DOI:** 10.3390/ijms24031873

**Published:** 2023-01-18

**Authors:** Andrea J. Detlefsen, Ryan D. Paulukinas, Trevor M. Penning

**Affiliations:** 1Department of Biochemistry & Biophysics, Perelman School of Medicine, University of Pennsylvania, Philadelphia, PA 19104, USA; 2Department of Systems Pharmacology & Translational Therapeutics, Perelman School of Medicine, University of Pennsylvania, Philadelphia, PA 19104, USA; 3Center of Excellence in Environmental Toxicology, Perelman School of Medicine, University of Pennsylvania, Philadelphia, PA 19104, USA

**Keywords:** aldo-keto reductase, bile acid, deficiency, congenital adrenal hyperplasia, hydroxysteroid dehydrogenase, prostate cancer, single nucleotide polymorphism, structure-function, steroid reductase, pseudohermaphroditism

## Abstract

Steroid hormones synchronize a variety of functions throughout all stages of life. Importantly, steroid hormone-transforming enzymes are ultimately responsible for the regulation of these potent signaling molecules. Germline mutations that cause dysfunction in these enzymes cause a variety of endocrine disorders. Mutations in *SRD5A2*, *HSD17B3*, and *HSD3B2* genes that lead to disordered sexual development, salt wasting, and other severe disorders provide a glimpse of the impacts of mutations in steroid hormone transforming enzymes. In a departure from these established examples, this review examines disease-associated germline coding mutations in steroid-transforming members of the human aldo-keto reductase (AKR) superfamily. We consider two main categories of missense mutations: those resulting from nonsynonymous single nucleotide polymorphisms (nsSNPs) and cases resulting from familial inherited base pair substitutions. We found mutations in human AKR1C genes that disrupt androgen metabolism, which can affect male sexual development and exacerbate prostate cancer and polycystic ovary syndrome (PCOS). Others may be disease causal in the AKR1D1 gene that is responsible for bile acid deficiency. However, given the extensive roles of AKRs in steroid metabolism, we predict that with expanding publicly available data and analysis tools, there is still much to be uncovered regarding germline AKR mutations in disease.

## 1. Introduction

Steroid hormones drive essential processes throughout all stages of life, including fetal gonadal differentiation, childhood and pre-adolescent development, adulthood, menopause, andropause, and cognitive decline. They regulate a diverse set of functions, including sexual differentiation, reproduction, water retention, electrolyte balance, and stress responses [1]. Underpinning this action are steroid hormone biosynthetic and metabolizing enzymes, which act as puppet masters and are responsible for the production and metabolism of these potent signaling molecules. However, when germline mutations cause dysfunction in these enzymes, the strings are cut, and a variety of endocrine-dependent disorders can ensue.

Over the years, there have been many well-documented cases of mutations in steroid hormone-transforming enzymes leading to a variety of disorders. Mutations in *SRD5A2* that result in the rare autosomal recessive 5α-reductase type 2 deficiency is a well-studied example. Normally, 5α-reductase type 2 converts Testosterone (T) to 5α-dihydrotestosterone (DHT), which is critical for male external genitalia differentiation and prostate development [2]. First documented in the 1970s in isolated clusters of individuals with 46,XY disordered sexual development (DSD), a biochemical catalog of mutations in *SRD5A2* and their impact on protein function and clinical presentation has expanded over the following decades [2]. Batista et al. provide a comprehensive evaluation of 451 identified cases to date of 5α-reductase type 2 deficiency in 48 countries, stemming from 151 allelic variants.

Relatedly, *HSD17B3* encodes for 17β-hydroxysteroid dehydrogenase type 3, which is highly expressed in the testes where it converts Δ^4^-androstene-3,17-dione (Δ^4^-AD) to T [3]. Mutations in this enzyme that lead to decreased T production underlie 17β-hydroxysteroid dehydrogenase type 3 deficiency, another rare autosomal recessive cause of 46,XY DSD. Due to a lack of T production in fetal testes, patients typically present at birth with a spectrum of phenotypes from completely female to ambiguous genitalia. However, individuals later experience amenorrhea and severe virilization during puberty [3]. A summary of clinical presentations, epidemiology, and demographic history of mutations causing 17β-hydroxysteroid dehydrogenase type 3 deficiency is detailed in a review from George et al. [3].

Mutations in the *HSD3B2* gene that result in 3β-hydroxysteroid dehydrogenase type 2 deficiency are one rare underlying cause of congenital adrenal hyperplasia (CAH) [4]. 3β-Hydroxysteroid dehydrogenase type 2 is responsible for the conversion of pregnenolone to progesterone, 17α-hydroxypregnenolone to 17α-hydroxyprogesterone, and dehydroepiandrosterone (DHEA) to Δ^4^-AD [4]. Consequently, its dysfunction in adrenal glands and gonads leads to DHEA accumulation, lowered cortisol and aldosterone levels, and dysregulation of other downstream and intermediate metabolites. Due to its extensive metabolic influence, the clinical presentation, diagnostic window, and phenotypic severity of 3β-hydroxysteroid dehydrogenase type 2 deficiency are heavily dependent on the location of the mutation within the gene and its outcome. Mutations that result in frameshift or protein truncation that completely eradicate enzyme activity cause severe salt-wasting that is diagnosed in the first weeks of life and is fatal if left untreated [5]. Missense mutations that leave some enzyme activity intact present as over-virilization in females and under-masculinization of males at birth. Even mutations that have a mild effect on enzyme activity can contribute to premature puberty, acne, and menstrual disorders in females, including polycystic ovary syndrome (PCOS) [5]. A detailed analysis of CAH caused by 3β-hydroxysteroid dehydrogenase type 2 deficiency can be found in reviews from Al Alawi et al. [4] and Sahakitrungruang [5].

Recently, Chang et al. detailed a unique scenario where a nonsynonymous single nucleotide polymorphism (nsSNP) in *HSD3B1* results in the gain of function N367T substitution that stabilizes its encoded protein, 3β-hydroxysteroid dehydrogenase type 1, leading to increased DHT production from adrenal-derived DHEA that could ultimately drive castration-resistant prostate cancer (CRPC) [6]. The authors describe a mode of selective adaptation in CRPC tumors that operates through loss-of-heterozygosity of the wild-type copy of *HSD3B1* in individuals who are germline heterozygous with wild-type/N367T alleles. This phenomenon leads to overexpression of the N367T allele due to increased resistance to ubiquitination and degradation, which allows for elevated intratumoral DHT production that can drive androgen receptor signaling and, ultimately, CRPC tumor growth [6]. In contrast to the previous examples where mutations were the source of severe disorders, the N367T nsSNP is an illustration of a mutant whose impact only becomes apparent when it exacerbates a disease.

The stories of *SRD5A2*, *HSD17B3*, and *HSD3B2* are flagship examples that establish the relevance of germline mutations in steroid metabolizing enzymes in disease. Each has been thoroughly documented, from mutation identification in patients to functional genomic studies that provide evidence of the protein dysfunction that is either disease causal or exacerbates disease. Comprehensive reviews that catalog known cases and consider what can be learned from the amassed data of each deficiency already exist and are referenced above. In a departure from these well-studied examples, this review will focus on the association of disease-associated germline coding mutations in members of the human aldo-keto reductase (AKR) superfamily that transform steroids. While there are many types of mutations that can lead to disease, our analysis considers two main categories: nsSNPs that results in a missense mutation and familial inherited single base pair transitions/transversions that result in rarer missense mutations. The main differentiation between these groups is that single nucleotide polymorphisms (SNPs) are classically defined as occurring in a significant portion of the population (minor allelic frequency >1%), while other damaging mutations are rarer. The reader is referred to Figure 1, which shows a generalized scheme for sex steroid hormone biosynthesis and metabolism and the genes that have germline mutations.

## 2. Aldo-Keto Reductases

AKRs are a superfamily of enzymes that reduce carbonyl substrates in an NAD(P) H-dependent reaction by a sequential ordered bi-bi kinetic mechanism [7]. Members share an (α/β)_8_-barrel motif with a highly conserved catalytic tetrad and cofactor binding site and possess three large loops (A, B, and C) at the back of the barrel, which defines substrate specificity [7]. Often these enzymes are pluripotent and can reduce 3-, 17-, and 20-ketosteroids as well as prostaglandins, retinol, or quinones. Several can also oxidize polycyclic aromatic hydrocarbon (PAH) *trans*-dihydrodiols [7]. Rather than discuss all 15 human AKRs, we focus on steroid-metabolizing enzymes, which account for much of the literature on disease-associated AKR mutations. Unlike the preceding examples, where a germline mutation results in a loss or gain of enzyme function that can be related to phenotypic change and disease outcome, the results are less complete for the AKRs. The one notable exception is the link between *AKR1D1* mutations and bile acid deficiency, which will be discussed later.

In our review of germline AKR mutations, we report three major approaches to identify possible links to disease. The first approach involves a survey of the epidemiological literature that identifies a collection of nsSNPs in a panel of AKR genes that may be associated with the disease. In the second approach, nsSNPs, without any formal link to disease, are identified, and biochemical characterization of the mutated protein is performed. The third approach identifies rare inherited mutations with an almost certain link to disease identified in the clinic and biochemically characterized. We found numerous reports claiming that nsSNPs in human *AKR* genes are associated with various aspects of disease without any functional genomic follow-up studies. In another case, the reverse was true, where analysis of several nsSNPs in *AKR1C2* uncovered kinetic differences in the enzyme that could theoretically impact prostate cancer (PCa). Still, no clinical study testing for this association was pursued. Even when a follow-up functional genomic study is conducted, it is possible that the effects of a deleterious mutation could be overlooked depending on whether the assay used is able to identify the impact of the mutation. For example, a study measuring steady-state kinetic parameters would not be suitable for detecting a change in protein stability. Furthermore, many diseases involving AKRs are polygenic, making it difficult to conduct studies that isolate the individual contributions of a single mutation. These efforts are further complicated due to AKRs’ involvement in multiple pathways of steroid metabolism where mutations in one step may be compensated for or compounded by nsSNPs in genes responsible for other steps in the same pathway. These obstacles in unraveling the roles of AKR mutations in polygenic diseases would ideally be overcome by studying them in vivo, where multiple mutations could be introduced by CRISPR/cas9. Unfortunately, the use of murine knockouts or transgenic mice to study AKR mutants is not feasible. Velica et al. [8] investigated eight of the nine existing murine AKR isoforms and found that they had largely different substrate and tissue distribution compared to human AKRs and, most importantly, that they were absent from many steroid hormone target tissues. Alternatively, CRISPR/cas9 could be used to create variants in cell models. The challenge of this approach lies in the design of specific guide RNAs, as the AKR1C genes share greater than 86% sequence identity.

We also consider that certain methodological assumptions may be made when searching for nsSNPs with a connection to the disease. Biases built into current systems, such as the origin of reference genomes, could, for example, tip the balance in determining which allele should be considered wild-type. As is evident from studies such as the 1000 genomes project, “wild-type” can shift between two alleles depending on which continental population is under consideration [9,10]. However, with the continually expanding publicly available data such as the recent additions to the UK biobank, the cancer genome atlas (TCGA), and more specific resources like the stand up to cancer (SU2C) prostate cancer foundation, these biases will likely dissipate [11,12,13]. This expansion also creates the potential for the categorization of mutations to be upgraded to an SNP status if the frequency in a newly evaluated population shifts the global minor allelic frequency (MAF). Additionally, programs that predict mutational impact in silico, such as PolyPhen and Sift, are useful in a first-pass evaluation of new mutations [14,15]. These programs provide a prediction of whether mutations are deleterious vs. benign, which often corresponds to amino acids that are evolutionarily conserved or non-conserved throughout the AKR superfamily respectively. Furthermore, the addition of new developments in computational analysis methods and tools like AlphaFold makes it increasingly more accessible to make an initial evaluation of the impact of mutations on structure in silico [16]. However, while these programs should be able to accurately predict the presence of the (α/β)_8_ barrel motif in mutant proteins, they may be challenged to predict the effect of mutations on the conformation of the disordered loops A, B, and C, which change upon ligand binding [17].

With help from advancements in available data and analysis tools, we predict that there is still much to be uncovered with respect to germline AKR mutations in disease. In addition to the examples detailed below, there is a web of connections linking AKRs to disease through their activity in key metabolic pathways, altered expression in female reproductive diseases such as endometriosis and PCOS, and general participation in a variety of other disorders and processes [18,19]. Given their wide reach, it is likely that there are other layers of genetic involvement for these enzymes that have yet to be discovered. Our goal is to provide a synopsis of germline coding mutations in steroid-transforming AKRs to reinvigorate the exploration of their connection to the disease.

## 3. Steroid Metabolizing AKR Enzymes

Members of the AKR1C subfamily display varying levels of 3-, 17-, and 20-ketosteroid reductase activities and metabolize several major classes of steroids. These reductive enzymes work in the opposite direction to oxidative hydroxysteroid dehydrogenases, where the complementary oxidoreductase activity of these enzyme pairs ultimately controls ligand availability and occupancy for steroid hormone receptors, including the androgen, estrogen, and progesterone receptors [18]. AKR1Cs also play a role in the metabolism of exogenous environmental toxicants that drive cancer initiation and progression. AKR1D1 is an exception and is the only steroid 5β-reductase in humans that catalyzes the 5β-reduction of the double-bond in Δ^4^-3-ketosteorids to produce 5β-dihydrosteroids, which are essential intermediates in bile acid biosynthesis. Importantly, several of these enzymes work in concert one with one another. For example, AKR1C3 is a major peripheral enzyme that converts Δ^4^-AD to T, which is a direct precursor to DHT. In contrast, AKR1C1 and AKR1C2 can metabolize DHT into the inactive 3β- and 3α-androstanediols, respectively. In another example, AKR1D1 precedes AKR1C4 in hepatic bile acid biosynthesis. Moreover, AKR1D1 works sequentially with AKR1C1 and AKR1C2 to provide a source of 3β,5β- and 3α,5β-tetrahydrosteroids, respectively. Genetic mutation of these critical enzymes may lead to disease at various stages of life. Understanding how these variants alter protein structure and function will improve the identification of at-risk populations and precision therapies. It is with this in mind that we chose to review current research and the remaining knowledge gaps in the study of germline mutations in human steroid metabolizing AKRs: AKR1C1-4 and AKR1D1.

### 3.1. AKR1C1

AKR1C1 (20α(3α)-hydroxysteroid dehydrogenase) is known to play an essential role in progesterone metabolism via its 20-ketosteroid reduction activity, where it converts progesterone to the inactive 20α-hydroxyprogesterone. Knockout of the murine homolog AKR1C18 leads to a delay in parturition due to the reduced ability to metabolize progesterone to 20α-hydroxyprogesterone [20]. The NCBI SNP database reveals no nsSNPs with missense outcomes that occur with an MAF greater than 0.01. A sequence alignment of the mutations detailed below and their locations with respect to conserved catalytic, cofactor, and substrate binding sites can be found in Table 1 and Figure 2. Recently a mutation in AKR1C1 was found to be associated with lipedema, a disease of subcutaneous fat accumulation [21]. In the context of inappropriate lipid accumulation, AKR1C1 is highly expressed in fat and liver tissues. Michelini et al. [21] hypothesized that an inherited AKR1C1 loss-of-function mutation could underpin nonsyndromic primary lipedema in one family, based on a similar accumulation of progesterone that promotes lipogenesis and lipid accumulation.

A 638 T > A transversion that results in a one amino acid substitution of L213 to Q in AKR1C1 was identified in a family with three members afflicted by sex-limited autosomal dominant nonsyndromic lipedema. Disease occurred in heterozygous carriers of the mutation. While the mutation did not affect protein expression in carriers, the L213Q variant was determined to be a loss-of-function mutation. L213 is not located on the active site. However, it is within the core of the protein, and its mutation appears to influence structure and function. Using molecular dynamic simulations, residue L213 was shown to participate in hydrophobic interactions. Mutation to glutamine disrupts these interactions due to the new polar side chain forming new hydrogen bonds. This mutation was found to affect the solvent accessibility of substrates with the steroid and cofactor binding pockets of AKR1C1. As a result of the structural changes, both steroid substrate and cofactor are more solvent-exposed, leading to a decrease in interaction energy between steroid substrate and cofactor that likely stems from a loss of non-covalent interactions. Quantitative structure-activity relationship (QSAR) modeling was performed to predict enzymatic parameters, where the mutant reduced both turnover number and catalytic efficiency by 50%.

This is a good example of the identification of a potentially clinically relevant mutation followed by the pursuit of functional genomic analysis to determine how it affects the protein of interest. However, the structure–function relationships were only performed with computational modeling and predictive relationships to determine reaction kinetics. These predictions were only performed with one steroid (20α-hydroxyprogesterone) and one cofactor (NADP^+^). The mutation may affect only one type of reaction, whereas conversion of other steroids may not be as drastically affected. Furthermore, the studies were not performed with the preferred substrates for AKR1C1, namely NADPH and progesterone. The QSAR modeling should have been ideally performed with additional steroid substrates to increase the robustness of the model’s predictions. Beyond progesterone conversion, the authors raise the idea that prostaglandin F 2alpha (PGF2α) is known to inhibit adipogenesis, whereas AKR1C1 catalyzes its synthesis [29,30]. Decreased AKR1C1 activity could release this brake on adipogenesis that is normally carried out by prostaglandins. However, we are unconvinced that the catalytic efficiency of this reaction would support this functional relationship since AKR1C3 appears to be the major PGF2a synthase [31]. The authors suggest that mutation of AKR1C1 could result in less PGF2α and stimulation of adipogenesis, which was described in their discussion. However, the group was unable to perform predictive kinetics on this reaction. These studies all point to the need to conduct kinetic analysis on each of the reactions of interest using recombinant enzymes bearing this mutation.

Lipid production is also known to be androgen-dependent in prostate cancer cells and PCOS adipocytes [32,33]. AKR1C1 inactivates the potent androgen receptor ligand DHT to 3β-androstanediol. Androgens have been shown to increase de novo lipogenesis and decrease lipid breakdown in adipocytes. Therefore, mutation of AKR1C1 would reduce DHT inactivation, allowing it to promote lipedema through androgen receptor signaling. Collectively, mutations in AKR1C1 may be able to promote lipedema through progesterone, androgen, or prostaglandin pathways. Therefore, it is important to perform additional functional genomic studies to determine how the mutation affects the reaction kinetics of progesterone conversion to 20α-hydroxyprogesterone, DHT to 3β-androstanediol, or prostaglandin E2 to PGF2α. These cases illustrate how mutations in a protein could have diverse effects that are ultimately mediated through the same pathway, from a delay in parturition in mice to dysregulated lipid accumulation in adults, both the result of decreased progesterone metabolism.

### 3.2. AKR1C2

AKR1C2 (type 3 3α-hydroxysteroid dehydrogenase) demonstrates preferred 3-ketosteroid reductase activity. Importantly, it inactivates the potent androgen DHT to 3α-androstanediol and converts dihydroprogesterone (DHP) to allopregnanolone, an important neurosteroid that modulates the GABA receptor. There are no k/o mice available since a murine equivalent of this enzyme does not exist [8]. The NCBI SNP database reveals one nsSNP with a missense outcome that occurs with an MAF greater than 0.01. It should be noted that several of the nsSNPs investigated by Takahashi et al. [34], detailed below, fall below the MAF threshold we used in this search. A sequence alignment of the mutations and nsSNPs is detailed below, and also their locations with respect to conserved catalytic, cofactor, and substrate binding sites can be found in Table 1 and Table 2, Figure 2. The impact of mutations in AKR1C2 on hormone-dependent diseases and disorders stems primarily from its involvement in DHT metabolism. As a potent androgen receptor ligand, DHT regulates sexual differentiation in embryonic males, promotes secondary sexual characteristics in adult males, and in some cases, causes androgen-dependent disorders. Mutations in AKR1C2 that result in an under or oversupply of DHT have been associated with several disorders.

Flück et al. described how several germline mutations in AKR1C2 underlie the dysregulation of male sexual differentiation observed in a Swiss family in 1972, beginning with the development of the fetal gonad driven by DHT [22]. Several members of a Swiss family exhibited varying degrees of under-virilization at birth, resulting in female sex assignment and 46,XY DSD [35]. Years later, Flück et al. sequenced DNA from the original family as well as one other family with similarly presenting individuals and identified a total of four germline inherited mutations in AKR1C2 that they further investigated for functional differences [22]. The authors proposed that fetal DHT synthesis proceeds through a backdoor pathway ending with the oxidation of 3α-androstanediol to DHT and that the identified mutations in AKR1C2 are the main cause of this pathway deficiency in affected individuals. They also identified a mutation in AKR1C4 that resulted in aberrant splicing, consequently eliminating its supplementary role to AKR1C2 metabolism and exacerbating the deficiency in the backdoor pathway to DHT. A kinetic analysis of wild-type and the three AKR1C2 variants identified in family one (I79V, H90Q, and N300T) showed reduced catalytic activity for two key reactions in the alternative pathway to DHT: the reduction of 5α-dihydroprogesterone (5α-DHP) to allopregnanolone, and the oxidation of 3α-androstanediol to DHT. The authors note that while AKR1C2 wild-type was able to oxidize 3α-androstanediol to DHT in vitro, it acts primarily as a 3-ketosteroid reductase in vivo based on its high affinity for NADPH. Therefore, the impact of these mutations most likely manifests in AKR1C2′s decreased ability to catalyze the reduction of 5α-DHP to allopregnanolone and 17α-hydroxyprogesterone to 17α-hydroxyallopregnanolone, both of which feed into the backdoor pathway [22], see Figure 3. Another AKR1C2 mutation, H222Q, was identified in the second family. In an assay performed in COS1 cells, the H222Q variant resulted in significantly reduced DHT production compared to wild-type [22]. The authors noted that while all four mutations identified in AKR1C2 resulted in reduced catalytic activity, it was not to the degree typically associated with recessive disorders of steroidogenesis. Ultimately, a combination of mutations in AKR1C2 and AKR1C4 is necessary to stunt fetal DHT synthesis and cause disordered sexual development in these two families.

In contrast to the previous study, where AKR1C2 mutations caused an undersupply of DHT that resulted in a sex development disorder, Takahashi et al. highlight how mutations in AKR1C2 that lead to an accumulation of DHT could exacerbate PCa. DHT is a potent AR ligand and a driver of PCa. As a mechanism to regulate its occupancy on the AR, DHT is reduced to 3α-androstanediol by AKR1C2, where disruption of this shunt could lead to an accumulation of DHT that exacerbates PCa [34]. Takahashi et al. examined how five AKR1C2 nsSNPs might affect protein function and disrupt DHT synthesis in PCa using transient transfection followed by enzyme assays in Sf9 cell lysates. They found that two variants, F46Y and L172Q, had decreased *V_max_* compared to wild-type in the reduction of DHT to 3α-androstanediol [34]. They also reported that three variants, L172Q, K185E, and R258C, had significantly lower apparent *K_m_* values compared to wild-type. None of the nsSNPs resulted in protein stability differences. The authors note that the frequency of the F46Y variant in several continental-based populations corresponds with the occurrence and severity of PCa in these groups. The F46Y allele is found in 15% of African and 5.9% of European populations and is undetected in Asian populations [34]. Interestingly, this mirrors the occurrence and severity of PCa in decreasing order for these ethnic groups. The authors highlight the parallel between F46Y occurrence and PCa and suggest that this or other AKR1C2 mutations could be a contributing factor to genetic PCa risk. They proposed that an association between the occurrence of the F46Y allele and elevated DHT serum levels in PCa patients may predict the severity of the disease; however, to our knowledge, this study has not been conducted. A weakness of the Takahasi study is that, as a control, they mutated the catalytic tyrosine and found that the Y55F mutant still had one-tenth the catalytic efficiency of wild-type AKR1C2. This calls into question the reliability of kinetic data obtained using Sf9 cell lysates as opposed to those obtained using purified recombinant mutant proteins.

In a follow-up study, Arthur et al. [36] used homology modeling to predict and analyze the structural impact of the mutations investigated by the Takahashi group. With respect to the F46Y variant, they report that the substitution of tyrosine introduces a polar residue into a previously hydrophobic environment. This tyrosine is predicted to form a hydrogen bond with a water molecule that could change the local environment and destabilize the hydrogen bond that normally forms nearby between N280 and the cofactor [36]. The authors suggest that this destabilization in cofactor binding could account for a reduced maximum velocity rather than F46Y directly disrupting the interaction with substrate DHT. They also report that the L172Q substitution might result in a similar disruption of hydrogen bonding between N167 and cofactor that leads to a decreased *V_max_* reported by Takahashi [35]. Together, these studies suggest that the F46Y nsSNP decreases AKR1C2′s catalytic ability due to disrupted cofactor binding, which could result in an accumulation of DHT. However, in the absence of a more rigorous kinetic analysis of the mutant proteins and a clinical study to confirm the impact of this finding in PCa patients, this example serves to highlight how a more comprehensive investigation spanning several disciplines is necessary to provide a complete story of the role of these nsSNPs in disease.

### 3.3. AKR1C3

AKR1C3 (type 2 3α(17β)-hydroxysteroid dehydrogenase) is the only steroid metabolizing AKR that preferentially displays 17-ketosteroid reductase activity that can convert Δ^4^-AD to T and 11-oxo-Δ^4^-AD to 11-keto-T [19]. It is also known as prostaglandin F2α synthase and oxidizes PAH-*trans*-dihydrodiols [31]. There are no k/o mice available since a murine equivalent of this enzyme does not exist [8]. The NCBI SNP database reveals five nsSNPs with missense outcomes that occur with an MAF greater than 0.01. A sequence alignment of the nsSNPs detailed below and their locations with respect to conserved catalytic, cofactor, and substrate binding sites can be found in Table 2, Figure 2. 

AKR1C3 drives prostate cancer and other hormone-dependent disorders, e.g., PCOS, where coding nsSNPs are thought to further modify their role in various aspects of the disease [37]. Until recently, most studies that implicated AKR1C3 nsSNPs were epidemiological and lacked functional genomic experiments to identify any change in protein function that could explain the connection to the disease. The AKR1C3 nsSNP rs12529, which corresponds to the H5Q mutation, has been flagged by many groups without further investigation. However, recent work from our group showed no major differences between AKR1C3 wild-type and the top four most frequently occurring variants, calling into question the weight of these associations.

Because of the role of AKR1C3 in the peripheral synthesis of T, there are numerous epidemiological studies that associate AKR1C3 nsSNPs, mainly H5Q, with PCa detection, prognosis, and treatment effectiveness. One study conducted in a New Zealand PCa cohort by Karunasinghe et al. suggested that the presence of the Q5 mutation in combination with smoking is associated with unusually low serum PSA levels that belie the severity of disease and lead to late detection by current prostate-specific antigen (PSA) diagnostic benchmarks [38]. The same group suggested that the Q5 mutation and smoking are associated with an increased age at which PCa is diagnosed by PSA serum levels [39]. Karunasinghe et al. also report that in patients receiving androgen deprivation therapy (ADT), the Q5 variant is associated with increased hormone treatment-related symptoms [40]. They proposed that these adverse drug effects could be avoided if individuals were genotyped for the Q5 variant to achieve precise treatment monitoring.

Three groups have associated AKR1C3 nsSNPs with deviations in serum T levels. Shiota et al. associated the presence of Q at position 5 with higher serum T levels in patients receiving ADT [41]. Inversely, the presence of H at position 5 is associated with a better prognosis. Relatedly, Jakobsson et al. described that in healthy Swedish subjects, the E77G variant is associated with lower serum T levels. However, they found no significant difference in enzyme activity of this variant compared to wild-type [42]. In the third study, Ju et al. report that Q5 is associated with increased serum T levels in a cohort of Chinese women with PCOS, which may indicate that the variant contributes to hyperandrogenism [43]. Notably, these three studies predict that mutations in AKR1C3 may affect T biosynthesis without performing any supporting biochemical characterization of the mutant proteins. Relatedly, several groups highlight AKR1C3 nsSNPs that might play a role in non-lethal disorders in men. Roberts et al. found that the Q5 mutation is associated with decreased risk of prostate enlargement in benign prostate hyperplasia [44]. Additionally, Soderhall et al. report the identification of a novel AKR1C3 nsSNP resulting in A215T substitution that is unique to a boy with penile hypospadias compared to a control group [45].

Studies implicating AKR1C3 nsSNPs extend beyond T production and PCa. The effects of AKR1C3 nsSNPs in the metabolic activation of PAHs and other carcinogens have also been considered. A preliminary study from one group suggests that the Q5 variant could be involved in the molecular pathogenesis of urinary bladder cancer [46]. In contrast, Figueroa et al. report an inverse association of Q5 with bladder cancer risk with no connection to smoking in a Caucasian population [47]. Interestingly, Lan et al. report the Q5 variant to be associated with a significant risk of developing lung cancer in a Chinese population exposed to high levels of PAH-rich coal combustions from cooking and heating [48]. Together, these studies highlight how AKR1C3 nsSNPs might modify their contribution to the production of ultimate carcinogens that can lead to cancer.

Given the large number of reports relating AKR1C3 variants with various positive and negative aspects of the disease, there are disproportionately fewer functional genomic studies examining how Q5 and other nsSNPs affect AKR1C3 function. To our knowledge, there are only two studies that biochemically characterize AKR1C3 nsSNPs. The first analysis, conducted by Platt et al., evaluated wild-type and five AKR1C3 variants in their ability to metabolize exemestane, an aromatase inhibitor used to treat breast cancer, to 17β-dihydroexemestane [49]. Their findings indicate that H5Q, E77G, K104D, P180S, and R258C are 17-250-fold less catalytically active compared to wild-type in this reaction, which could significantly affect exemestane metabolism in breast cancer patients with different variants and warrant the consideration of AKR1C3 genotype in treatment protocols [49]. However, in our own work, where we evaluated wild-type and the top four most frequently occurring variants (H5Q, K104D, E77G, and R258C), we found no significant kinetic differences in the ability of the variants to metabolize Δ^4^-AD to T, progesterone to 20α-hydroxyprogesterone, or exemestane to 17β-dihydroexemestane compared to wild-type [10]. Additionally, while the K104D variant was less stable than WT, the presence of cofactors NAD(P)^+^ diminished this effect. In contrast to associative and biochemical studies that implicate H5/Q5 and other AKR1C3 nsSNPs in disease, our findings indicate that none of the variants we examined have significant differences that would likely manifest in patient prognosis.

While several of the associative studies discussed above convey contradictory impacts of the H5Q variant on disease prognosis, the sizable number of reports that flag this variant makes it difficult to excuse these associations completely. However, our own work is in opposition to the idea that H5Q is an influential variant, as we found no significant impact of this nsSNP or any other on AKR1C3 protein function or stability. H5Q is the most commonly occurring AKR1C3 variant with a global MAF of 0.42 [9,10]. However, in certain continental populations, the distribution is reversed, and Q5 is the major allele. This raises the concept that depending on the population observed in a given study, Q5 could be the major allele in contrast to the global MAF. This could lead to a distorted enrichment of the Q5 variant in certain populations. Furthermore, other factors, such as the upregulation of AKR1C3 via ADT or other modifications of its expression that occur in PCa and other disease states, could obscure the influence of the H5Q variant and account for the widespread identification of this variant in associative studies.

### 3.4. AKR1C4

AKR1C4 (type 1 3α-hydroxysteroid dehydrogenase) is predominantly expressed in the liver, where it displays 3-ketosteroid reductase activity and is responsible for making 3α-hydroxysteroids. However, alteration of AKR1C4 has been associated with mood disorders. Progesterone is converted into an intermediate (5α-dihydroprogesterone) by SRD5A1, which is then available for conversion by AKR1C4 into allopregnanolone, which is implicated in negative mood changes due to dysregulation of GABAergic signaling of glutamatergic neurons [50]. However, it is thought that the main AKR involved in central nervous system (CNS) regulation via allopregnanolone metabolism is AKR1C2 [51]. There are no k/o mice available since a murine equivalent of this enzyme does not exist [8]. The NCBI SNP database reveals six nsSNPs with missense outcomes that occur with an MAF greater than 0.01. A sequence alignment of the mutations and nsSNPs detailed below and their locations with respect to conserved catalytic, cofactor, and substrate binding sites can be found in Table 2, Figure 2.

One study identified the C145S AKR1C4 nsSNP due to a C to G transversion in a purely associative study [52]. The majority of patients in this study had type 1 bipolar disorder and exhibited an irritable mood during mania/hypomania based on affective disorders evaluation (ADE). The SNP was associated with increased manic or hypomanic irritability, where men with the SNP were 5.44 times more likely to experience manic or hypomanic irritability compared to those without the SNP. This effect was not seen in women. Paradoxically, as increased irritability correlated with the C145S variant in men, there was a corresponding decrease in serum progesterone levels. The authors speculated why this could occur but did not follow up with any mechanistic studies. The same group published follow-up studies in another population of men and women with bipolar disorder [53]. There was a correlation between men with paranoid ideation and both DHEA-S and progesterone levels, where the mean levels of both steroids were lower in men with paranoid ideation compared to those without. In terms of the C145S variant in AKR1C4, women had a reduced likelihood of exhibiting paranoid ideation, indicating that the mutation may have the reverse outcome and yield a protective effect in women. However, in order to determine whether the C145S mutation causes a change in AKR1C4 catalytic activity that reflects decreases in DHEA-S and progesterone levels, it would be necessary to conduct a functional genomic analysis of the protein. It is also uncertain whether these changes would be related to changes in systemic steroid metabolism in the liver or CNS. For example, the NCBI database only shows transcript expression in the liver and gall bladder.

AKR1C4 was also linked to breast cancer in a population of postmenopausal women who were receiving estrogen or combined estrogen and progesterone therapy [54]. The group observed that carriers who were heterozygous or homozygous for L311V SNP correlated with a 16.7 and 29.3% increase, respectively, in mammographic percentage density (MPD), a risk factor for breast cancer in women who were receiving combined estrogen and progesterone as hormone replacement therapy. However, the sample size was small for these groups, with an N of seven and one, respectively, so these findings should be considered with caution. It was previously reported that L311V does have a 66–80% decrease in AKR1C4 catalytic activity, so this residue may be important for substrate binding since it resides in the C-terminal loop [55]. Since functional data about the mutant was already known, the group was able to relate this SNP to the MPD risk factor in postmenopausal women who were receiving combined estrogen and progesterone replacement therapy.

### 3.5. AKR1D1

AKR1D1 (Δ^4^-3-oxosteroid-5β-reductase) is a key enzyme for bile acid synthesis, and disruption of this critical pathway leads to bile acid deficiency. AKR1D1 reduces Δ^4^-cholesten-7-ol-3-one and Δ^4^-cholesten-7,12-diol-3-one to their respective 5β-dihydrosteroid forms. AKR1D1 works sequentially with AKR1C4 to produce the 3α,5β-configuration in the A-ring of the steroid, which is essential for the proper emulsification of fats, Figure 4. Interestingly, steroid 5b-reductase k/o mice retained some ability to synthesize bile acids. Still, the bile acid levels were reduced, and composition differed in males and females, where the former had significantly reduced 12a-hydroxylated bile acids [56]. If undetected in humans, bile acid deficiency can be a fatal disorder in the neonate due to the resulting inability to emulsify fat and absorb fat-soluble vitamins. In addition, these mutations lead to diversion in the metabolism of 7α-hydroxy-4-cholesten-3-one and 7α,12α-dihydroxy-4-cholesten-3-one to the 5α-reduced (allo)-bile acids which are hepatotoxic. Historically the treatment of this deficiency focused on relieving symptoms rather than identifying the underlying genetic causes of the disorder, leading to the possibility that the frequency of this genetic disorder may be underestimated. A sequence alignment of the mutations detailed below and their locations with respect to conserved catalytic, cofactor, and substrate binding sites can be found in Table 1 and Figure 2.

There are several mutations in AKR1D1 associated with bile acid deficiency. Lemonde identified two transitions in infants with cholestatic liver disease [23]. One patient was homozygous for a 662 C > T transition that resulted in a P198L substitution, while their parents were both heterozygous for the mutation. This mutation was not found in a control population of 100 chromosomes. The group also identified a second mutation (385 C > T), resulting in an L106F substitution where the patient was similarly homozygous for the mutation and had parents that were both heterozygous, Table 1.

Gonzales et al. described a case report where a pair of 8-month-old sisters with progressive cholestasis and liver failure were compound heterozygous for two missense mutations in AKR1D1: P133R (467 C > G) in exon 4 and R261C (850 C > T) in exon 7 [25]. Each parent was heterozygous for one of the mutations, and it was confirmed that both girls had two mutated alleles. Two patients with chronic cholestasis had a heterozygous mutation resulting in a G223E (737 G > A) amino acid substitution in exon 6 [28]. Finally, another case report identified two infants with neonatal cholestasis [27]. One was heterozygous for a novel R266Q mutation, which was detected in the heterozygous mother but not in the father. The second patient was compound heterozygous for the G223E and R261C mutations. G223E was detected as heterozygous in the patient’s father and absent in the mother, while R261C was heterozygous in the patient’s mother and absent in the father. Recently, another group identified more inborn errors of bile acid metabolism in three infants [26]. One patient had a compound mutation consisting of R50X, where X indicates an early stop codon and the R266Q mutation. The second had a chromosomal mutation (74 C > T), resulting in a T25I mutation. These studies present the connection between AKR1D1 deficiency and its effect on bile acid metabolism and the development of cholestasis. However, it is necessary to evaluate relevant biochemical studies on the structure–function relationship to determine why these mutations result in bile acid deficiency.

Our group conducted an extensive functional genomic analysis of how these mutations affect AKR1D1 structure and function. The point mutations L106F, P133R, P198L, G223E, and R261C were tested to determine how they affect AKR1D1 [24]. All mutations are highly conserved across AKR1D1 homologs in other mammalian species except for P133R. None of the mutations were in direct contact with the catalytic tetrad, cofactor, or substrate binding sites. Interestingly, only P133R could be successfully purified while the other four accumulated in inclusion bodies indicating protein aggregation, misfolding, or instability. G223E degraded within 24 h in cycloheximide pulse-chase experiments conducted in transfected HEK293 cells, while L106F and R261C were poorly expressed and degraded within 6 h. 5β-Reduction of T was assessed for each mutant, and very low activity was seen with L106F and R261C within 24 h, which is consistent with their poor expression. Additionally, no conversion was observed with the G223E or P198L mutations over 60 h. However, as reaction times increased, background conversion of T to other androgens occurred, making it difficult to quantify residual 5β-reduction of T. Protein expression was measured for each of the five mutations, revealing that L106F, R261, and G223E all had reduced stability. P198L remained in cells for up to 24 h; however, the mutation may still affect its enzymatic activity or expression since it could not be successfully purified.

The P133R mutation was the only recombinant enzyme to be successfully purified, and reactions could be conducted to determine steady-state kinetic parameters. When using T as a substrate, the *K_m_* increased from 2.7 to 12.7 µM, and the *k_cat_* decreased from 7.1 to 2.7, resulting in an over 10-fold decrease in the catalytic efficiency. When using cortisone as a substrate, the *K_m_* decreased from 15.1 to 1.3 µM. However, the *k_cat_* also decreased from 9.9 to 0.6, ultimately resulting in only a small reduction in the catalytic efficiency. This illustrates how kinetic parameters for two different substrates could be drastically different for the same mutation in an enzyme. The substrate of interest Δ^4^-cholesten-7α-ol-3-one has a longer C17 side chain, similar to that of cortisone. However, the *K_m_* for this substrate could not be accurately measured due to saturation at the lowest substrate concentration. At saturation, *K_m_* was significantly lower than 0.8 µM (the value for wild-type), while the *k_cat_* was reduced 7-fold. These differences were more like those observed with cortisone and reframe AKR1D1 from a low affinity, high-capacity enzyme to a high affinity, low-capacity enzyme in this scenario. This could indicate that AKR1D1 binds tightly to bile acid substrates resulting in insufficient turnover to the 5β-reduced bile acid precursors for proper emulsification, which could account for bile acid deficiency from this mutant.

Thermal stability studies showed that 50% of mutant enzymatic activity was lost at 42 °C and less than 5–10% remained at 46.5 °C, while >60% of the enzymatic activity of wild-type remained. This indicated that the instability of the mutants is exacerbated at temperatures above 37 °C. Thus, while the mutants are less stable than the wild-type enzyme, this observation is not relevant at physiological temperatures, and it is thought that differences in kinetic parameters may be more important.

Transient kinetics were performed with the P133R mutant, where it was shown to result in a 40-fold increase in the *K_d_* value for NADPH and an increased rate of release of NADP^+^ by two orders of magnitude compared to wild-type. The reduced affinity for the cofactor suggests that the enzyme exists in a cofactor-free form. Impaired NADPH binding and hydride transfer were found to be the molecular basis for bile acid deficiency in patients with the P133R mutation [57].

Here we see that biochemical studies become critical to understand how a point mutation may affect structure and function. If only evaluated by computational methods, these mutations may have been overlooked as they are largely not near the catalytic tetrad, substrate, or cofactor binding sites. However, the biochemical studies brought to light why these mutations may lead to bile acid deficiency due to disruption of AKR1D1 expression/stability or catalytic activity. However, each point mutation can have a different effect on enzyme function, and the same mutation, P133R, can have different effects on different steroid substrates that can only be revealed by transient kinetic studies. Molecular dynamic studies could have been useful in understanding how these nsSNPs may directly affect protein stability, as we saw with the AKR1C1 analysis. The biochemical studies support the association of the AKR1D1 mutants with cholestasis/liver failure and bile acid metabolism deficiency. It was suspected that AKR1D1 might be the offender; however, the biochemical studies aid in definitively explaining why these nsSNPs alter the structure and function of AKR1D1.

## 4. Conclusions

AKR steroid metabolizing enzymes are critical players in proper physiological growth and development. Germline mutations in these enzymes can disrupt androgen, progesterone, and bile acid metabolism and lead to debilitating pathological disease states that greatly diminish the quality of life and could ultimately lead to death, especially in young individuals born with these variants. These coding-region point mutations ultimately alter the proper expression levels and function of these key metabolizing enzymes. In many cases, epidemiological studies link a variant to a specific disease state; however, the functional genomic studies to support the association and understand the structure and functional impact of the mutation are often incomplete.

Most often, a patient is treated for a phenotype. However, there may be different underlying dysfunctions that lead to the same disorder. This is evident in the examples highlighted in the introduction, where mutations in *SRD5A2*, *HSD17B3*, and *HSD3B2* all lead to some form of disordered sexual development. Only with a close examination of which key steroid metabolites were in excess in patients in combination with sequencing can these disorders be teased apart. In rare cases, when the diagnosis is based on a phenotype without investigation of the underlying cause of the disease, a patient could even be misdiagnosed. This is where precision medicine is helpful in recognizing that patients with the same categorical disease may need different types of treatment. Additionally, mutations in a protein should be considered in the greater context of the entire metabolic pathway of which the enzyme is a part. This is especially true in the case of human AKRs, which typically catalyze several reactions in alternative routes to the same end steroid. If one portion of the pathway is shunted due to a mutation that hinders the biological activity of the enzyme, this may drive the pathway through another arm. Disruption of this metabolic pathway could even lead to the formation of novel steroids, making it important to study the steroid metabolome in the context of a mutation to understand how the disease state arises. The accumulation of SNPs in the same pathway should also be considered, as the combination of mutations may have debilitating cumulative effects.

When examining the impact of a SNP, it could also be important to consider how a mutation may precipitate disease in men and women differently. As can be seen in the proteins highlighted in the introduction, certain deficiencies cause severe phenotypes in one gender over the other due to the nature of steroid signaling. This is also evident in the case of AKR1C3, where T production can affect men and women differently through PCa and PCOS, respectively. Therefore, mutations may have different outcomes depending on the role of the enzyme’s target product in different sex-dependent developmental and reproductive processes.

With the recent additions to large genome databases such as the UK biobank and increasing accessibility to analytical tools like AlphaFold, there are new opportunities to unearth connections between AKR SNPs and disease. It will be essential to draw from both associative and biochemical studies in order to uncover the full story of a mutation and evaluate how this knowledge may be incorporated into medical intervention. As is evident from the diversity of research groups and the expanse of time in some of the more developed stories, this work often requires perseverance by many different groups with complementary expertise and resources. With this perspective, we might consider some of these narratives pending rather than permanently incomplete.

## Figures and Tables

**Figure 1 ijms-24-01873-f001:**
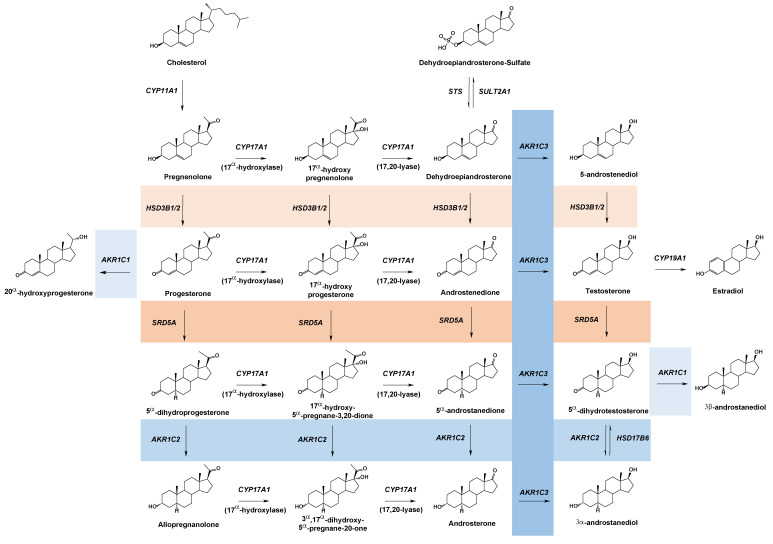
Human sex steroid hormone metabolism with short-chain dehydrogenases/reductase enzymatic conversions influenced by germ-line mutations. AKR family members reviewed in the present work are shaded in blue, while non-AKR family members previously reviewed are shaded in orange.

**Figure 2 ijms-24-01873-f002:**
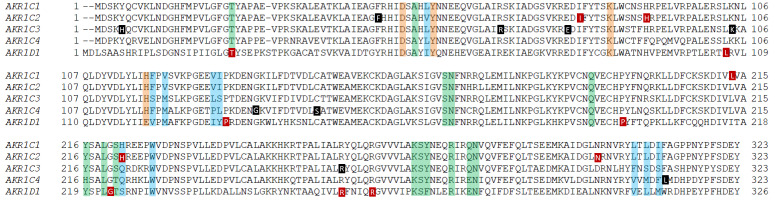
AKR sequence alignment shows the position of major nsSNPs and inherited mutations. Catalytic tetrad is shown in orange, steroid binding residues in blue, cofactor binding residues in green, nsSNPs with an MAF greater than 0.01 in black boxes, and familial mutations in red boxes.

**Figure 3 ijms-24-01873-f003:**
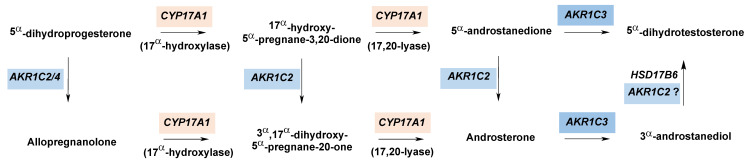
Role of AKR1C2 and AKR1C4 in the fetal backdoor pathway for DHT. Arrows depict reactions carried out by the indicated enzymes. The involvement of AKR1C2 is unconfirmed (indicated by a question mark) in the reaction converting 3α-androstanediol to 5α-dihydrotestosterone.

**Figure 4 ijms-24-01873-f004:**
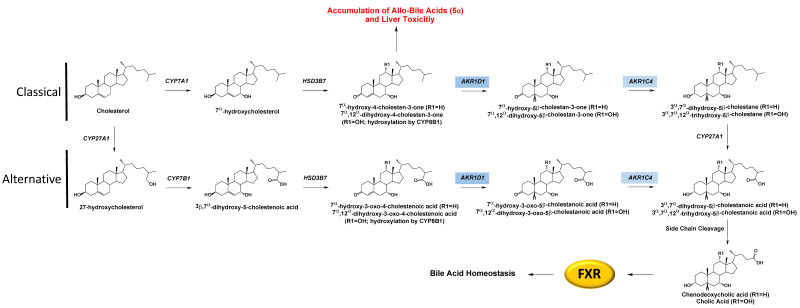
Role of AKR1C4 and AKR1D1 in Classical (neutral) and Alternative (acidic) Bile Acid Metabolism to regulate farnesoid X receptor (FXR). The FXR is required to regulate bile acid homeostasis. 5β-reductase deficiency, such as mutation of AKR1D1, shunts the pathway away from proper bile acid metabolism and leads to the accumulation of allo-bile acids resulting in liver toxicity.

**Table 1 ijms-24-01873-t001:** Familial inherited missense mutations in steroid transforming human AKRs.

AKR Enzyme	Missense Mutation	References	Disorder
AKR1C1	L213Q	Michelini et al., 2020 [21]	Nonsyndromic Primary Lipedema
AKR1C2	I79V	Flück et al., 2011 [22]	46,XY DSD
	H90Q		
	N300T		
	H222Q		
AKR1D1	P198L	Lemonde et al., 2003, Drury et al., 2010 [23,24]	Bile Acid Deficiency
	L106F		
	P133R	Gonzales et al., 2004, Drury et al. 2010, Chen et al., 2020 [24,25,26]	
	R261C	Gonzales et al., 2004, Seki et al., 2013, Drury et al., 2010 [24,25,27]	
	G223E	Ueki et al., 2009, Seki et al., 2013, Drury et al., 2010 [24,27,28]	
	R266Q	Seki et al., 2013, Chen et al., 2020 [26,27]	
	T25I	Chen et al., 2020 [26]	

**Table 2 ijms-24-01873-t002:** Steroid-transforming human AKR nsSNPs with an MAF greater than 0.01, * indicates that nsSNPs with a frequency of <0.01 exist but are not included in the table.

AKR Enzyme	Missense nsSNP	NCBI Identifier	MAF
AKR1C1 *	-	-	-
AKR1C2	F46Y	rs2854482	0.0649
AKR1C3	H5Q	rs12529	0.4203
	K104D	rs12387	0.1518
	E77G	rs11551177	0.0367
	R258C	rs62621365	0.0325
	R66Q	rs35961894	0.0230
AKR1C4	C145S	rs3829125	0.1028
	L311V	rs17134592	0.1024
	G135E	rs11253043	0.0270
AKR1D1 *	-	-	-

## Data Availability

No new data were created or analyzed in this study. Data sharing is not applicable to this article.

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
