# Peer review of "Germline Mutations in Steroid Metabolizing Enzymes: A Focus on Steroid Transforming Aldo-Keto Reductases"

_ijms, 2023, doi:10.3390/ijms24031873_

Round 1

Reviewer 1 Report

Here, the authors have provided a review that examines the pathophysiological involvement of AKR1 enzymes in steroid metabolism. Although the same authors recently produced a comprehensive review on the same topic in Endocrine Review 2019, they decided here to focus more on germline variants found in patients, and discuss to which extent non-synonymous single nucleotide polymorphisms (nsSNPs) versus variants found in familial cases, are informative in deciphering the physiological contribution of AKR1 in steroid hormone and bile acid homeostasis. The review is well written and properly updated given the short time between this review and the very comprehensive Endocrine Review.

General comment: In discussing the relevance of associating AKR1 variants with phenotypes of altered steroid metabolism, the authors' argument essentially oscillates between 2 points of view, that of the geneticist/epidemiologist and that of the biochemist. Between these two extremes, the use of genetic models in mice is often an excellent way to determine the causal link between a putative loss-of-function or gain-of-function variant and the reality of in vivo conditions. Of course, as the authors explain very well, the multiple connections and possible overlap between metabolic pathways can make interpretation difficult, but this is also what is expected to happen in patients. In addition, the ability to target the genetic alteration in the chosen tissue using conditional alleles also avoids possible interactions between different organs involved in steroid or bile acid metabolism. Similarly, the Crispr/cas9 approach can be very useful to reproduce variants found in patients or to consider modifying several family members simultaneously based on their highly conserved domains. Although some AKR1C isoforms do not appear to have orthologs in mice, their function may be retained but taken over by another AKR?  It would be helpful to readers if the authors included comments on these different points in their review.

Specific comment: It should be informative to include a Table recapitulating kinetic parameters of AKR1 variants compared to “normal” AKR1 from the different teams having performed functional enzymatic studies?

Minor comments:

Line 148: spell out MAF as it is the first mention

Line 157: I guess you mean “conformation”?

Line 195: “y” was lacking in 20a-hydroxyprogesterone

Line 196: AKR1C18 instead of C8

Figure 3: its seems that this is the alternative backdoor pathway for DHT production, maybe it should be written in the legend?

Line  348: only 5 nsSNPs were described in figure 2 and not 6 as mentioned in the text

Line 350: Table 2 instead of Table 1

Line 443: only Table 2

Line 495: Table 1 instead of Table 2

Line 507:Table 1 instead of Table 2

Author Response

Response to Reviewer 1 comments for ijms-2055755

We would like to take this opportunity to thank the editors and reviewers for their constructive critique and by answering their concerns we believe that the manuscript is improved. We will now respond to the critiques in the order in which they appear.

Reviewer #1: Here, the authors have provided a review that examines the pathophysiological involvement of AKR1 enzymes in steroid metabolism. Although the same authors recently produced a comprehensive review on the same topic in Endocrine Review 2019, they decided here to focus more on germline variants found in patients and discuss to which extent non-synonymous single nucleotide polymorphisms (nsSNPs) versus variants found in familial cases, are informative in deciphering the physiological contribution of AKR1 in steroid hormone and bile acid homeostasis. The review is well written and properly updated given the short time between this review and the very comprehensive Endocrine Review.

We thank the reviewer for this positive comment.

General Comment:

  1. In discussing the relevance of associating AKR1 variants with phenotypes of altered steroid metabolism, the authors' argument essentially oscillates between 2 points of view, that of the geneticist/epidemiologist and that of the biochemist. Between these two extremes, the use of genetic models in mice is often an excellent way to determine the causal link between a putative loss-of-function or gain-of-function variant and the reality of in vivo conditions. Of course, as the authors explain very well, the multiple connections and possible overlap between metabolic pathways can make interpretation difficult, but this is also what is expected to happen in patients. In addition, the ability to target the genetic alteration in the chosen tissue using conditional alleles also avoids possible interactions between different organs involved in steroid or bile acid metabolism. Similarly, the Crispr/cas9 approach can be very useful to reproduce variants found in patients or to consider modifying several family members simultaneously based on their highly conserved domains. Although some AKR1C isoforms do not appear to have orthologs in mice, their function may be retained but taken over by another AKR?  It would be helpful to readers if the authors included comments on these different points in their review.

We thank the reviewer for their insightful comments. First, we agree that we have described both epidemiological associations of genetic variants with health and disease but have also described the functional genomic studies to determine whether they could underpin the associations described. This is a necessary first step to determine causality. In the revised manuscript we have emphasized the difference between epidemiological findings and functional genomics studies, for example in lines 126-132. This perspective is also reflected in conclusion lines 652-657. We also make the point that functional genomic studies can miss the effect of a deleterious mutation depending on the assay used lines 137-141. For example a stability mutant may not be detected in steady state kinetic parameters.

Second, we agree that loss-of-function or gain-of-function mutations in AKR1C enzymes would be best modeled in an in vivo setting. However, it is not possible to model the effects of mutations in the AKR1C human genes in mice lines 146-154. Our AKR web-site identifies four human AKR1C genes but nine in mice (see https://akrsuperfamily.org/). The article by the Bunce group set out to determine functional murine orthlogs of the human genes. They cloned and expressed eight of the nine murine isoforms and found that they differed in substrate specificity and tissue distribution, and importantly that they were absent from many steroid hormone target tissues (see Velica et al., Mol. Cancer 2009, 8:121). We respectfully conclude that using murine k/o or transgenic mice to examine loss or gain of function mutants is not technically feasible. We elaborate on this point in the manuscript. Regarding the reviewer’s final point, the possibility that in mice the role of one human AKR1C gene may be taken over by another is not supported by the Bunce group work.

Third, we agree that a Crispr/cas9 approach would be useful to reproduce the variants in cell models. We agree that this approach could be used to delete an AKR1C gene to show the effect of a loss-of function mutation and could be used for gene editing to create a desired mutation. As the reviewer rightfully points out this approach could lead to modifying several AKR1C genes at once since they have >86% sequence identity. Thus, the challenge is in the design of the guide RNA. We now make this point in the revised manuscript in lines 152-154.

Specific comment:

  1. It should be informative to include a Table recapitulating kinetic parameters of AKR1 variants compared to “normal” AKR1 from the different teams having performed functional enzymatic studies?

We thank the reviewer for this thoughtful suggestion. However, the heterogeneous nature of the type of kinetic parameter reported in each study, the substrate(s) used, and the quality and source of the enzyme would make it difficult to make comparisons across studies. For example, investigation of AKR1C3 variants by Platt et. al. reported mainly CLint for the aromatase inhibitor exemestane and not kcat. They found significant differences in nsSNP variants compared to WT. However, in our own investigation of AKR1C3 nsSNPs by Detlefsen et. al. [ref 9] we observed minimal differences in the same variants in the metabolism of exemestane and for three additional steroid substrates. Furthermore, Drury et. al  [ref 56] was only able to report kinetic parameters compared to WT for the P133R AKR1D1 mutant, as it was the only protein able to be purified. The other AKR1D1 mutants rather reflected significant changes in stability. Michelini et. al. reported kinetic parameters for the L213Q AKR1C1 mutant based on in silico computational analysis only. Finally, the investigation of the F46Y AKR1C2 nsSNP by Takahashi et. al. was performed in Sf9 cell lysates, where the catalytically dead Y55F mutant that was used as a control still showed some activity. This calls into question the reliability of their kinetic data, and highlights the danger of using impure enzyme preparations to draw conclusions. It is with this in mind that we have chosen not to include a table with these parameters.

Minor comments:

  1. Line 148: spell out MAF as it is the first mention

We have corrected this, now in line 165.

  1. Line 157: I guess you mean “conformation”?

We have corrected this mistake, now in line 175.

  1. Line 195: “y” was lacking in 20a-hydroxyprogesterone.

We have corrected this, now in line 215.

  1. Line 196: AKR1C18 instead of C8

We have corrected this, now in line 216.

  1. Figure 3: its seems that this is the alternative backdoor pathway for DHT production, maybe it should be written in the legend?

We have clarified this and included “backdoor” in the figure legend.

  1. Line  348: only 5 nsSNPs were described in figure 2 and not 6 as mentioned in the text

This statement has been corrected to “five”, now in line 374.

  1. Line 350: Table 2 instead of Table 1

We have corrected this, now in line 376.

  1. Line 443: only Table 2

We have corrected this error, now in line 470.

  1. Line 495: Table 1 instead of Table 2

We have corrected this error, now in line 522.

  1. Line 507:Table 1 instead of Table 2

We have corrected this error, now in line 534.

Reviewer 2 Report

In this manuscript entitled “Germline mutation in steroid metabolizing enzymes: a focus on steroid transforming aldo-keto reductases”, the authors summarize their findings regarding mutations in the AKR1 subfamily involved in the disease. The manuscript concise explanations of the advantages and disadvantages of different studies.

Comments:

1. Are AKR1C4 and AKR1D1 mutation sites found outside of the germline? Or are there known germline phenotypes due to these mutations? The author should reconsider the title of the review to something better.

2. Why do the authors focus on AKR1? Based on the analysis of various variants such as nsSNPs, understanding the world of AKR1 is crucial for our quality of life. The authors should clearly write an Introduction chapter on the need to address AKR1 in this review. 

3. The authors need to add a table summarizing the relationship between disease and missense nsSNP mutations in AKR1. Table 2 could be used.

4. Figures 1~4 are not effectively inserted in the text and the authors need to reconsider the text. The figures themselves are good.

5. Lines 138-158 are methodological and authors should make them as short as possible.

6. Why does a mutation in a different site of AKR1C1 result in a completely different phenotype? And is it possible that a secondary phenotype may emerge in which the inability to sustain a gestation is due to defective lipid production? The authors should address this question.

7. There is a mistake in Table 2. Also, are there any nsSNP missense mutations in AKR1D1? If so, is Table 2 in the AKR1D1 section an error in Table 1?

8. Abbreviations should be written when they first appear.

Author Response

Response to Reviewer 2 comments for ijms-2055755

We would like to take this opportunity to thank the editors and reviewers for their constructive critique and by answering their concerns we believe that the manuscript is improved. We will now respond to the critiques in the order in which they appear.

Reviewer #2: In this manuscript entitled “Germline mutation in steroid metabolizing enzymes: a focus on steroid transforming aldo-keto reductases”, the authors summarize their findings regarding mutations in the AKR1 subfamily involved in the disease. The manuscript concise explanations of the advantages and disadvantages of different studies.

General comments:

  1. Are AKR1C4 and AKR1D1 mutation sites found outside of the germline? Or are there known germline phenotypes due to these mutations? The author should reconsider the title of the review to something better.

The scope of this review covers mutations and variants that are passed from parent to offspring (both rare familial mutations and more widely occurring nsSNPs), as opposed to somatic mutations that occur after conception or are acquired as a result of selection during disease development. We discuss germline mutations in both AKR1C4 (section 3.4: missense nsSNPs associated with aspects of mood disorders and breast cancer) and AKR1D1 (section 3.5: rare familial mutations resulting in bile acid deficiency). While it is possible that there are also somatic mutations in these enzymes, we feel that the title reflects the scope of this discussion of these two enzymes as well as the others.

  1. Why do the authors focus on AKR1? Based on the analysis of various variants such as nsSNPs, understanding the world of AKR1 is crucial for our quality of life. The authors should clearly write an Introduction chapter on the need to address AKR1 in this review.

We have added additional text in lines 205-207 to further emphasize why we chose to focus on human steroid-metabolizing AKRs that is detailed in the “Steroid Metabolizing AKR Enzymes” section of the introduction. We chose to omit discussion on two enzymes in the AKR1 family that are outside the scope of androgen metabolism: AKR1A1, an aldehyde reductase and AKR1B1, an aldose reductase which play pivotal roles in the inactivation of reactive aldehydes and the polyol pathway.

  1. The authors need to add a table summarizing the relationship between disease and missense nsSNP mutations in AKR1. Table 2 could be used.

The majority of the nsSNPs in Table 2 have not been shown to result in biochemical differences and are not explicitly associated with a disease. We chose to include them based on their MAF being greater than 0.01, since their prevalence will be detected in population-based genetic studies. Such studies may in the future associate a variant with a disease, which could then be validated by conducting functional genomic studies. Many epidemiological studies, such as those referenced in the AKR1C3 section 3.3, identify an initial association between a mutation and an aspect of disease, but do not pursue biochemical validation as to whether a mutation leads to loss or gain of function in the protein. By contrast, the mutations associated with disease in Table 1 have been shown to be causal. It is with this in mind that we have decided to keep the data in the two Tables separate.

  1. Figures 1~4 are not effectively inserted in the text and the authors need to reconsider the text. The figures themselves are good.

The figures in the document were compiled by the journal. We will work with the journal to ensure that the final figure placement is optimal.

  1. Lines 138-158 are methodological and authors should make them as short as possible.

We respectfully disagree with this comment. The purpose of lines 138-158 (now lines 155-176) is to bring attention to recent advancements in publicly available tools that could be leveraged to identify new associations between steroid metabolizing human AKRs and disease. This section also serves to highlight possible pitfalls of this approach. This section is critical because as ‘big-data” accumulates it will not be feasible to conduct functional analysis on each variant and these tools provide a mechanism by which predictions can be based and functional analysis can be triaged. 

  1. Why does a mutation in a different site of AKR1C1 result in a completely different phenotype? And is it possible that a secondary phenotype may emerge in which the inability to sustain a gestation is due to defective lipid production? The authors should address this question.

We appreciate this comment and have clarified the language in lines 215-217, 223-226, 235-237,  and 273-276 to address these questions. The L213Q mutation in AKR1C1 associated with nonsyndromic lipedema ultimately results in a reduced ability to convert progesterone to 20α-hydroxyprogesterone. The authors hypothesize that elevated progesterone could increase lipogenesis and lipid accumulation. They also note that AKR1C1 is involved in prostaglandin (PG) F2α formation, which normally inhibits adipogenesis. Decreased AKR1C1 activity could release this brake on adipogenesis that is normally mediated by prostaglandins. However AKR1C3 appears to be the major human PGF2α synthase. On the other hand, AKR1C18 k/o mice have a defect in progesterone metabolism. AKR1C18 is an ortholog of  human AKR1C1, and AKR1C18 is the only known murine homolog of a human AKR1C gene. The phenotype of these mice is as expected in that they are unable to initiate parturition due to a reduced ability to metabolize progesterone to 20α-hydroxyprogesterone. It is possible that this deficiency could also affect lipid accumulation in the maternal mice, but this was not explored in the study. However, Michelini et. al. cites that progesterone has a lipogenic effect on adipose tissue in mice, mediated through the sterol regulatory element-binding protein 1c (SREBP1c). In conclusion, both the deficiency of the murine ortholog and the L213Q mutation in AKR1C1 results in phenotypes that are derived from the inability to metabolize progesterone.

  1. There is a mistake in Table 2. Also, are there any nsSNP missense mutations in AKR1D1? If so, is Table 2 in the AKR1D1 section an error in Table 1?

We are unsure what the mistake in Table 2 is. There are no AKR1D1 nsSNPs that result in missense mutations that occur at a MAF greater than 0.01, the criteria we used to construct Table 2. We now place an asterisk next to AKR1D1 to indicate that nsSNPs with a frequency less than 0.01 exist but these are not listed.

  1. Abbreviations should be written when they first appear.

We have ensured that abbreviations appear when they are first written.

Round 2

Reviewer 2 Report

  1. Lines 138-158 are methodological and authors should make them as short as possible.

 We respectfully disagree with this comment. The purpose of lines 138-158 (now lines 155-176) is to bring attention to recent advancements in publicly available tools that could be leveraged to identify new associations between steroid metabolizing human AKRs and disease. This section also serves to highlight possible pitfalls of this approach. This section is critical because as ‘big-data” accumulates it will not be feasible to conduct functional analysis on each variant and these tools provide a mechanism by which predictions can be based and functional analysis can be triaged.

(Reviewer) If included, it should be emphasized with references that methodological developments are important and can contribute to the understanding of steroid hormone metabolism such as AKRs. Furthermore, it should be emphasized that it is an essential tool for future disease research.

  1. There is a mistake in Table 2. Also, are there any nsSNP missense mutations in AKR1D1? If so, is Table 2 in the AKR1D1 section an error in Table 1?

We are unsure what the mistake in Table 2 is. There are no AKR1D1 nsSNPs that result in missense mutations that occur at a MAF greater than 0.01, the criteria we used to construct Table 2. We now place an asterisk next to AKR1D1 to indicate that nsSNPs with a frequency less than 0.01 exist but these are not listed.

(Reviewer) Is the missense nsSNP site S145C (Table 2) the same as C145S (line 481)? If both are the same, no correction is needed.

Author Response

Response to Reviewer 2 comments for ijms-2055755-1

1. Lines 138-158 are methodological and authors should make them as short as possible.

We respectfully disagree with this comment. The purpose of lines 138-158 (now lines 155-176) is to bring attention to recent advancements in publicly available tools that could be leveraged to identify new associations between steroid metabolizing human AKRs and disease. This section also serves to highlight possible pitfalls of this approach. This section is critical because as ‘big-data” accumulates it will not be feasible to conduct functional analysis on each variant and these tools provide a mechanism by which predictions can be based and functional analysis can be triaged.

Reviewer comment: If included, it should be emphasized with references that methodological developments are important and can contribute to the understanding of steroid hormone metabolism such as AKRs. Furthermore, it should be emphasized that it is an essential tool for future disease research.

We thank the reviewer for their comment. Tools including PolyPhen, Sift, and now AlphaFold, are widely used by structural biologists to assess structure function relationships. We forecast that improvements in underlying artificial intelligence technologies of these tools in combination with the avalanche of new genomic data will lead to advancements in the field of disease associated mutations in human steroid metabolizing AKRs. We acknowledge that this is our opinion, and we believe that an essential part of writing a review is to use our expertise to provide an assessment of upcoming areas of growth in the field. An example of such progress in the related short chain dehydrogenase/reductase (SDR) superfamily can be found in the following paper by Persson et. al. (PMID: 19027726), where a functional subdivision of the SRD superfamily was divided into at least 200 SDR families based on hidden Markov models. In our review we also acknowledge the potential short comings of these approaches when applied to loop structures: “while these programs should be able to accurately predict the presence of the (α/β)8 barrel motif in mutant proteins, they may be challenged to predict the effect of mutations on the conformation of the disordered loops A, B, and C which change upon ligand binding”. We therefore provide a balanced description of the utility of these advances and respectfully suggest that no changes in the manuscript are required at this time.

2. There is a mistake in Table 2. Also, are there any nsSNP missense mutations in AKR1D1? If so, is Table 2 in the AKR1D1 section an error in Table 1?

We are unsure what the mistake in Table 2 is. There are no AKR1D1 nsSNPs that result in missense mutations that occur at a MAF greater than 0.01, the criteria we used to construct Table 2. We now place an asterisk next to AKR1D1 to indicate that nsSNPs with a frequency less than 0.01 exist but these are not listed. 

Reviewer comment: Is the missense nsSNP site S145C (Table 2) the same as C145S (line 481)? If both are the same, no correction is needed.

We thank the reviewer for bringing this error to our attention. We have corrected the error in table 2.
